# Hospital Staff Report It Is Not Burnout, but a Normal Stress Reaction to an Uncongenial Work Environment: Findings from a Qualitative Study

**DOI:** 10.3390/ijerph17114107

**Published:** 2020-06-09

**Authors:** Madeleine Kendrick, Kevin Kendrick, Peter Morton, Nicholas F. Taylor, Sandra G. Leggat

**Affiliations:** 1School of Psychology and Public Health, La Trobe University, Bundoora VIC 3086, Australia; 19460000@students.latrobe.edu.au; 2WA Health, Perth WA 6004, Australia; kevin.kendrick@health.wa.gov.au (K.K.); peter.morton@health.wa.gov.au (P.M.); 3School of Allied Health, Human Services and Sport, La Trobe University, Bundoora VIC 3086, Australia; n.taylor@latrobe.edu.au; 4School of Public Health, Harbin Medical University, Harbin 150081, China

**Keywords:** burnout, stress, occupational health, work

## Abstract

(1) Background: The issue of burnout in healthcare staff is frequently discussed in relation to occupational health. In this paper, we report healthcare staff experiences of stress and burnout. (2) Methods: In total, 72 healthcare staff were interviewed from psychiatry, surgery, and emergency departments at an Australian public health service. The sample included doctors, nurses, allied health professionals, administrators, and front-line managers. Interview transcripts were thematically analyzed, with participant experiences interpreted against descriptors of burnout in Maslach’s Burnout Inventory and the International Statistical Classification of Diseases and Related Health Problems (ICD-11). (3) Results: Staff experiences closely matched the ICD-11 description of stress associated with working in an uncongenial workplace, with few reported experiences which matched the ICD-11 descriptors of burnout. (4) Conclusion: Uncongenial workplaces in public health services contribute to healthcare staff stress. While previous approaches have focused on biomedical assistance for individuals, our findings suggest that occupational health approaches to addressing health care staff stress need greater focus on the workplace as a social determinant of health. This finding is significant as organizational remedies to uncongenial stress are quite different from remedies to burnout.

## 1. Introduction

Since authors Freudenberger and Maslach [1,2] published their works on burnout, studies on how ‘unmitigated, chronic stress’ impacts individuals have evolved from identifying antecedents to discussing remedies and preventative measures. Burnout in healthcare is reported with increasing frequency [3,4,5]. The consequences of burned-out healthcare staff are staff shortages and lower quality of patient care [6,7].

Research identifying how ‘burnout’ can overlap with other conditions, such as depression [8], has suggested the need to further distinguish burnout from other workplace-related issues to improve worker health and wellbeing. To assist with this differentiation the authors took a different approach to most studies of burnout. This article explores patient-facing healthcare staff perspectives of stress and burnout by using the International Statistical Classification of Diseases and Related Health Problems, 11th edition (ICD-11) mental health diagnostic codes to augment Maslach’s Burnout Inventory (MBI) as the theoretical framework of stress, burnout, depression, and anxiety. Depression and anxiety were included as the psychiatric literature discusses potential overlaps and commonalities between burnout and other mental illnesses [8,9].

Maslach has described burnout as involving, “…Emotional exhaustion, depersonalization, and lack of personal accomplishment” (p. 192). The ICD-11 described burnout as “a syndrome conceptualized as resulting from chronic workplace stress that has not been successfully managed” [10]. The ICD-11, a diagnostic aid published by the World Health Organization (WHO), records these criteria for use in daily clinical practice. The ICD-11 outlines specific mental or behavioral conditions, such as generalized anxiety disorder, depression, and acute and chronic disorders associated with stress [10]. In addition, the ICD-11 includes differential classifications for ‘problems related to employment and unemployment’ that can impact health and wellbeing. One such problem, listed under the code QD83.0, is the ‘problem associated with uncongenial work’. The issue of ‘uncongenial work’ crosses the spheres of public health and organizational psychology, as specific conditions in the workplace are well established in the literature as a contributing factor to burnout [11,12].

The aim of the research was to explore staff experiences of stress and burnout in a large health service. This research question was developed based on literature, which has highlighted the link between a challenging work environment and high rates of burned-out staff among, e.g., urgent care doctors [4,13] and nurses [14]. Surgical and emergency professions are well-known for difficult working environments, which routinely include negative interpersonal interactions, shift work at unsocial hours, and the high acuity of the work [13,15]. However, the existing research targeting health professionals is largely quantitative [16], aiming to link various personal (e.g., emotional intelligence, job satisfaction) and organizational factors (e.g., leadership, social support) to reported burnout. This study aimed to fill a gap by exploring perceptions of stress and burnout in depth to ensure these constructs were clearly understood before antecedents and solutions were identified.

## 2. Materials and Methods

### 2.1. Study Design and Setting

The design of this study involved qualitative analysis of a series of in-depth semi-structured interviews to explore staff perceptions of their workplace experiences. Interview questions were designed involving consultation with project stakeholders on the research focus. Researchers then collected data independently of stakeholder input. No data were collected during the research design stage of the project, only suggestions. The study was set in two urban hospitals in a single public health service in Australia located in one major city. This allowed data to be combined for analysis. Three clinical departments were sampled. Staff from psychiatry, surgery, and emergency departments provided a range of perspectives from clinical areas known for high acuity work.

### 2.2. Study Population

The participants included in the frontline ‘patient-facing’ staff include doctors, nurses, allied health professionals, administration and clerical staff, and front-line managers (such as nurse-managers who work on the ward). Patient-facing support services staff were excluded from the study because in one hospital these workers were employed by a private agency and not the health service directly. Participants were recruited using a combination of purposive and snowball sampling, where the researchers aimed to sample an entire team of patient-facing staff in the target departments. The process of recruitment involved an initial purposive approach using email introductions with the participant information statement attached that outlined the aims and protocol of the study, and in-person explanations of the study given pre-interview to confirm understanding. From the purposive approach, targeting staff in the psychiatry, emergency, and surgical departments, individuals were encouraged to tell their co-workers about the study and encourage others to participate. Individuals in the target departments were encouraged to discuss the study and the option of participating regardless of whether they had participated themselves.

### 2.3. Procedures

The study took a constructivist epistemology to the creation of knowledge from the subjective experiences and perceptions of individuals [17]. In line with this exploratory approach, the researchers used open-ended interviewing. Recommended by stakeholders during the project design stage, the key question from which responses were analyzed for this paper was, ‘Please describe what work is like for you (?)’. Follow-up questions depended on the response given to the first question, including prompts such as, ‘What causes stress for you when you’re at work?’ and ‘what helps you to de-stress after work?’. Questions about the participant’s experience and understanding of the concept of burnout included, ‘What does the term ‘burnout’ mean to you?’ and, ‘Have you ever experienced burnout? (If so), What did it feel like for you?’.

Due to the semi-structured, qualitative nature of the interview, participant responses varied based on their experiences and priorities. This approach was informed by studies that have illustrated the use of shorter questionnaires and analytical tools to identify the presence of burnout [18,19], deriving from the themes in the MBI [2]. The researchers followed the example of authors such as West, Dyrbye et al. [18] with a simplified set of reflective questions based on MBI. A qualitative research design was chosen to enable in-depth exploration of participant views to address the research question.

Interviews were conducted by two interviewers. One interviewer was a doctoral research student (MK), who designed the project, with a background in management interviewing techniques. The other interviewer was a psychiatric registrar, with a background in psychiatric interviewing techniques (KK). The two interviewers collaborated during transcription and coding to ensure overall consistency and to discuss assumptions prior to analysis. When participants from the psychiatry department were familiar with the psychiatrist interviewer, the doctoral research student conducted the interview to maintain objectivity, and ensure that all responses could be kept confidential. In other cases, participants felt more comfortable speaking to the psychiatrist interviewer. Interviews were conducted face-to-face with the interviewer with audio recording for accurate transcription. The researchers also recorded participant responses with rich descriptions, due to the nuances communicated by non-verbal communication such as extended pauses, sarcasm, and emphasis on specific words. Three participants declined to be recorded during the interview and their responses were written as field notes by the researcher. 

Interviews averaged approximately 40 min in duration, in a private location at the participant’s discretion (such as in an office, or a consultation room after-hours). Participants volunteered knowing that there was no compensation. All interview transcripts were de-identified during transcription for participant confidentiality. Each participant was given the opportunity to review and edit their resulting transcript; the few transcripts which were returned to the researchers had only minor corrections.

### 2.4. Data Management and Analysis

Data analysis was completed similar to steps presented in Braun and Clarke [20], in the form of deductive thematic analysis. Audio recordings were transcribed and re-reviewed. This was followed by generation of preliminary codes. From these codes overarching themes were identified for analysis. A two-level analysis was then conducted, initially of coded data extracts, and then of the whole data set with iterative recoding as necessary. Based on this, themes and subthemes were defined, considering preexisting descriptions of themes and terminology in academic literature. The completed transcripts, coding and collation was managed using NVivo qualitative data analysis software (QSR International Pty Ltd. Version 12, 2018, Melbourne, Australia).

The purpose of deductive thematic analysis was to identify themes and trends in the data relevant to the project and theoretical framework [21] based on the collected experiences and perceptions of participants [17]. The two interviewers completed coding, with the doctoral student coding the majority of transcripts, and the psychiatrist interviewer cross-coding a small number to ensure consistency as recommended by the COREQ checklist [22]. A further check was provided with another academic management member of the research team (SL) reviewing a sample of the de-identified transcripts and codes. This approach aimed to minimize researcher bias through coding by researchers from different professional backgrounds and documented discussion between team members to encourage transparency and self-reflexivity [21,22,23,24,25]. 

The process used to code documents was deductive and iterative, highlighting key words in the transcripts when they matched the ICD-11 theoretical framework of stress, burnout, depression, and anxiety. In the case of ‘anxiety’, key words such as ‘anxious’, ‘worried’, and ‘nervous’ would be highlighted and assigned to the main theme. When key words were not enough to qualify an anecdote (such as participant describing feeling ‘nervous’ and ‘stressed’), the context of the anecdote and the participant’s wider interview were considered before assigning a theme. The final assignation was based on ICD-11 guidelines with oversight from the consultant psychiatrist clinical site supervisor (PM). The transcripts were re-read and coded with a wider scope, incorporating quotes and anecdotes made by participants in relation to themes from the framework.

### 2.5. Ethics

The project was approved by the Health Service and La Trobe University Human Research Ethics Committees (HREC), registration number PRN: 0966. After receiving a participant information and consent form, all participants provided written informed consent prior to participating in the study. A withdrawal of consent form was provided to all participants in case they wished to no longer participate in the project. No participants withdrew their information. The research followed the consolidated criteria for reporting qualitative studies (COREQ) [22].

De-identifying and collectively organizing participant transcripts was an essential requirement of the ethics approval. Without the ability to protect the identities of participants, the research project would not have been allowed to proceed due to the negative impacts that identification could have on participants. Variables such as age, gender, and years of service were divulged by participants, but were not recorded to preserve confidentiality. In addition to de-identifying individual transcripts, the researchers stored individual consent forms separately in a secure location in the researchers’ private office. Given the confidential nature of the data, the transcripts cannot be made available in open access.

## 3. Results

A total of 72 participants were interviewed, with 35 from psychiatry, 7 from surgery, 22 from emergency, and 8 from ‘other’ categories, such as those who worked across both surgical and emergency departments. Participants included doctors, nurses, and allied health professionals, as well as administration staff and line managers. (See Table 1).

In the psychiatric and emergency departments leaders (such as senior nurse-managers, and administration personnel) openly endorsed participation to their colleagues, encouraging volunteers to contact the researchers. As a result, in emergency and psychiatry, the researchers were able to sample several complete teams of patient-facing staff (three teams in psychiatry, two teams in emergency). Data from the surgery and ‘other’ departments were included for analysis due to thematically consistent responses. While direct representation of a population in the data sample is required for statistical analysis, the distinctive advantage of semi-structured interviews is that a theme can reach saturation from a comparatively small cohort [21,22,23,24,25]. This phenomenon was experienced by researchers; the outcomes of which are discussed in the rest of the article.

When participants were asked to describe ‘what work was like for them’, the majority of participants discussed experiences of stress, anxiety, and burnout. While many individuals used the term ‘burnout’, when asked to describe what burnout meant to them, only a small number of participants described burnout as the “end stage of chronic stress” defined by Maslach [2] and the ICD-11. Very few individuals described neither workplace stress, anxiety, nor burnout.

### 3.1. Burnout or Something Else

When data collection was conducted and participant responses were compared with themes found in MBI and contemporary literature on burnout [11,16], a major theme emerged. Participants often felt that they or their co-workers were ‘burned out’, yet most participant descriptions (68 of 72) of experiences of ‘burnout’ were, in fact, not consistent with clinical indicators of burnout [10].

#### Differentiating Burnout from Other Conditions

To illustrate the differences between participant anecdotes that fit the descriptions of burnout and those which did not, two examples from the data have been selected. Participant 4’s responses to interview questions matched the diagnostic and research criteria for burnout. Participant 40’s did not, matching more closely to descriptions of stress.
“Got to a point…and that’s only been in the last few years…but it’s got to a point now where I was like…I can’t feel this miserable every day and be…a good person. Do a good job and do all the things I do like everyone else. … Been a bit up-and-down, to be honest…. I know it sounds ridiculous, being a [clinician] and everything, but you know…in my head, I was like “I was fine! Until this place happened.” And now, this place has happened. And now, you know, that’s it. I would say the only time I’ve got back to brilliant is when I’m not here. So, when I’m on holiday. And I’ve had two big holidays while I’ve been here. Thankfully my [period of leave] came in, so I had [a lot of time] off work. It was the best. I didn’t want to come back, obviously”.(participant 4)
“…they were dropping me into some night shifts…and I was having palpitations. I could feel it there…like oh god I’m dreading…I was really dreading going in **participant pats their chest during the anecdote*.* Going online and [obsessively] checking the [high] patient numbers. That’s what I started doing! But that’s not good for me. Like, **recounting, visibly anxious** oh my god, I’m going into this”.[which subsided once the participant began their shift] (participant 40)

Participant 40 described their experiences as ‘much better’ once they began their work routines; they felt more at-ease and less stressed. Participant 40 proceeded to describe how much they love their work, and how satisfying it was to connect with their patients. Participant 40 continued going to work because, “I still love my job! I love socializing, talking with patients…”, and because they derive a sense of deep satisfaction from providing high-quality, attentive care to patients. Like participant 40, most participants, including those who acknowledged significant stress, did not describe themselves or their co-workers as being ‘emotionally exhausted’, ‘depersonalized’, or ‘lacking feelings of personal accomplishment’. The lack of descriptors of this nature indicated to researchers that these experiences were not burnout. Instead, participants regularly described experiences of consistent, daily stress as a consequence of their working environment, consistent with data from organizational psychology research on the negative impact of uncongenial working environments [9,26]. In short; the participant’s reactions to the stressors of their working environment were not within the parameters of burnout disorders described by the ICD-11 or Maslach [2]. Instead, it was the workplace that was ‘disordered’. These employees were working within the uncongenial workplace and were exhibiting reasonable reactions to an unreasonable working environment.

Participant 4’s experience differed from participant 40’s in many ways. Participant 40 appeared to experience anxiety, but this was largely anticipatory, as it evaporated once work began. In contrast participant 4 experienced anxiety and unhappiness that continued throughout work. They were also unable to detach from these sensations during weekends away from work, remaining focused on their experience of misery. These sensations only eased during prolonged holiday periods. This is congruent with evidence that one of the few known remedies for burnout discussed in literature is time away from the workplace where those suffering from burnout usually experience complete recovery from their symptoms [12,27]. This in and of itself would not have been sufficient to qualify as likely burnout, but outside of the anecdote, the participant also described emotional exhaustion, depersonalization, loss of personal accomplishment, professional cynicism, and a loss of professional efficacy, which are all criteria identified in Maslach’s work [2]. On review of the case by the Consultant psychiatric supervisor, participant 4 was identified as one of the few participants describing burnout, whereas participant 40, whose experience of anxiety was largely limited to checking patient numbers prior to the shift, did not ruminate on work was not classified as burnout. The experiences of participant 40 were far more common within our sample than those of participant 4.

### 3.2. The Uncongenial Workplace and Its Impact on Staff Wellbeing

Participants who appeared to experience the most stress in the workplace were staff in the middle of the patient-facing hierarchy, i.e., those who were neither junior, nor senior staff in the team. These staff included registrar doctors and registered nurses. As their role carried a large amount of responsibility, often with little autonomy, these ‘middle’ staff members explained that, “There is a lot of responsibility in the workplace, a lot of stress…” (participant 12), and, “Why is this my responsibility to do …I brought [it] up with every single level of person… but it always used to come back to me, it was horrendous.” (participant 4). Many of the participants listed other stressors that impacted their long-term wellbeing, such as feeling unsafe at work, feeling like there was more work to do than people available to do it, and user-unfriendly systems for forms and issue resolution.
“…it’s almost like…you want to get something done for the benefit of patients and staff…and you’re happy to go to all the effort to prove why it’s needed and everything…but then it’s the continual…’oh, nobody knows’ or ‘you’ll have to speak to so-and-so somewhere up the ladder about that’, and you hit loads of dead-ends just trying to find out some very basic information” **participant rolls their eyes, emphasizing the word ‘basic’**.(participant 72)

When asked to describe how these experiences made them feel, participants volunteered ‘fatigue’ and ‘frustration’ most frequently. Participants expanded on this, explaining that their frustrations mostly stemmed from systems and situations which prevented them from performing their job to their best ability;
“…I got a classic phone call the other day **irritated*…*There was a position being abolished [unnecessarily, as the position is actually required in the organization]… but there’s no money to reinstate the position now that it’s been cut, so the workload has to be absorbed by current staff. There’s no ‘fat’ in the system to absorb extra work anymore... So, we are all being provisioned to provide our services at a basic level, when we want to deliver silver or gold-standard care! When we can’t do that, that’s when the dissatisfaction comes in. **participant emphasizes the word ‘dissatisfaction’*.* When you know you’re not delivering that standard, and there’s somebody out there hurting because you can’t, I feel concern for them...**participant looks upset by this disp-quote*”*.(participant 17)

This was the main theme, with interview participants describing feeling stressed when they believed that they were not providing optimal quality of patient care. Those staff in the middle of the hierarchy, were more likely to describe this stress than more senior or junior staff.

## 4. Discussion

The researchers sought to explore the patient-facing staff experiences of stress and burnout in a large health service by comparing interview responses to MBI [2] and the ICD-11 [9]. The data of this exploratory study and its implications came as a surprise after the focus of burnout in media and academic sources suggested a trend in public health workplaces [3,8,12]. Our data revealed two themes. The first suggested that despite increasing awareness, the concept of burnout is largely misunderstood and is frequently co-opted to describe experiences of long-term physical fatigue, chronic stress, and anxiety [10,28] experienced in an uncongenial workplace.

The second theme suggested that stress was most common among patient-facing staff in the middle of the hierarchy, and this was largely a consequence of their perceived inability to provide the quality of care they felt they should be providing. These findings are consistent with current research that has found a relationship between episodes of missed care and powerlessness and stress among nurses [29].

These themes led to the conclusion that the experiences of those interviewed was not burnout, but stress arising from uncongenial working conditions. Despite participants self-reporting that they perceived their experiences to be ‘burnout’, their own descriptions did not match the ICD-11 nor Maslach’s [2] descriptions of the burnout condition. Instead, participant anecdotes and descriptions more closely matched with ICD-11 descriptions of stress and anxiety related to an uncongenial workplace. The study’s data and analysis were presented to the research team’s consultant psychiatrist (PM), who agreed with this perspective.

### 4.1. Occupational Health and Staff Stress

The results of this study illustrated a case where participants suggested that they and their colleagues were suffering from burnout, but which in fact were stress responses to workplace factors, as also identified by Murtagh [30]. Despite the increasing attention on burnout of healthcare workers, recent systematic reviews have found limited understanding of the prevalence [31] and the factors contributing to burnout [32], making it difficult to build the evidence on effective interventions [29,33]. Whether the impacts on health are physical or psychological, it is the responsibility of a workplace to create an environment that is safe for an individual to conduct their work, with minimal risk [34,35,36]. This responsibility requires the workplace to take initiative to ensure staff safety, from training and education to regular risk assessments to check that safety standards are maintained [34].

While physical safety standards are well documented [37], the same cannot be said for psychological safety at work [38], such as the means for creating congenial workplaces to minimize staff experiences of stress [39]. The field of burnout and staff psychological safety in the workplace is still in development, as research primarily focuses on measurements of stress and burnout, as well as individualistic remedies, such as personal resilience capacity [40]. We suggest the next step in the field’s evolution involves implementation studies and interventions to assess the applicability of theoretical findings such as those discussed by Lamontagne, Keegel and Vallance [36] and Shain, Arnold and Germann [41].

### 4.2. Implications for Public Health

While our analysis showed that the majority of participants in this project did not suffer from burnout, burnout remains the endpoint of long-term stressful experiences for an individual [1,2,10]. Therefore, if left to continue, an uncongenial working environment with large numbers of stressed healthcare staff is likely to contribute to burnout in the future. A further consideration is the negative effect on patient care by a stressed healthcare workforce [42]. What this means for public health policymakers, government departments, and hospital boards, is a need to recognize and address the conditions an uncongenial workplace creates for staff, particularly those working in high-acuity clinical professions. Remedies may include audits and service improvement initiatives addressing staff access to information systems [43], and leadership responses to staff concerns in the workplace to better support employees [44]. Access to information and leadership support for patient-care duties was a significant concern of our participants.

Rather than developing strategies to bolster individual staff resilience to these uncongenial working conditions, a more sustainable and effective solution may be to remedy specific stressors present in the workplace. This will require accurate assessment of social factors and psychological stressors, in addition to physical workplace factors, in each public hospital for which policymakers and boards are responsible. To this end, the process of a staff-focused service review is highly recommended for the immediate future.

### 4.3. Limitations

Our results may be inconsistent with current literature on burnout due to targeting an infrequently sampled population or sampling a relatively small percentage of the overall number of staff working at the target health system. For context, the staff employed at the target health system number in the thousands, crossing a multitude of medical specialties, of which 72 participants were interviewed from three departments. However, by the end of the interviews no new themes emerged suggesting that data saturation had been achieved. There is potential for the participants of this study to experience work differently to those who have participated in other studies due to a variety of unique conditions present in this study’s target health system, some of which were identified in Kendrick and Bartram et al. [45]. Further research may be required to confirm whether the results are generalizable to other health settings.

Due to the field of burnout being relatively new with diagnostic and measurement tools still undergoing revision and improvement [16], there is potential for burnout to be mis-diagnosed by clinicians and academics alike in the search for a solution to stress from an uncongenial workplace. This interpretation raises a question for future studies on burnout rates in healthcare staff: what percentage of ‘burned-out’ individuals are genuinely burnt-out and what percentage are simply in need of a more functional place in which to conduct their work? The use of the participants’ own description of their perceptions of stress and burnout offer a unique insight into the lived experiences of staff. This nonetheless presented a potential limitation of responder bias, which was suggested by the high number of participants responding that they felt burned-out, forcing the study to rely on cross-referencing the work of Maslach [2] and the ICD-11 [10] to qualify these anecdotes using established theory. Future research might also benefit from including additional variables, such as respondent gender and age, to draw further conclusions.

## 5. Conclusions

Burnout was rarely present despite participants self-reporting feeling burnt-out, or ‘witnessing’ burnout in their co-workers. This finding diverged from current literature on burnout in healthcare workers, which suggested a trend of high burnout rates across a range of cultural contexts [3,4,18]. The stress reaction to an uncongenial workplace reported by our study participants was largely attributed to staff inability to provide the quality of care they felt was necessary to best serve their patients but did not typically reach the criteria for burnout as defined by either Maslach or the ICD-11.

This unexpected finding and its interpretation led the authors to three potential conclusions: either (a) current studies on burnout are inaccurate and thus skewed towards confirmation-bias; (b) the target sample is a significant outlier that requires in-depth investigation as to the reasons why participant experiences do not match current trends of burnout; or (c) qualitative study has enabled greater depth in understanding the experiences of the staff that are not possible with existing quantitative burnout questionnaires [17,21,23]. This finding is also significant as organizational remedies to uncongenial stress are both scarcer [46] and different from remedies to burnout [47]. Healthcare organizations should pay greater attention to the working environment, taking care to ensure that hospitals are supportive of patient-facing staff duties and personal wellbeing. This study’s results provide a rationale for shifting the future focus of health practitioner wellbeing away from individual factors, such as personal resilience, towards organizational factors that act as social determinants of health.

## Figures and Tables

**Table 1 ijerph-17-04107-t001:** Participant professional and clinical specialty groups.

	Doctors	Nurses	Admin	Allied Health	Line Management	Total
Psychiatry	7	6	7	9	6	35
Surgery	1	4	0	2	0	7
Emergency	7	7	3	1	4	22
‘Other’	0	1	2	0	5	8
Total	15	18	12	12	15	72

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
