# Peer review of "Hospital Staff Report It Is Not Burnout, but a Normal Stress Reaction to an Uncongenial Work Environment: Findings from a Qualitative Study"

_ijerph, 2020, doi:10.3390/ijerph17114107_

Round 1
Reviewer 1 Report
INTRODUCTION:
First sentence: missing citation # 1.
The last line of para 1 is confusing. Please revise.
There is background about burnout and it would be beneficial for the readers if there was some description of stress in health service or employment (exisiting research). It will help to differentiate between stress and burnout.
MATERIALS AND METHODS
This is a well-written section. I have a few comments and queries as follows:
2.3 Procedures - how long (on average) were the interviews? When (weeknds, breaks, during work hours, etc?) and where were they conducted? Were participants compensated for their time?
2.4. Data management and analysis, Line 116 - The authors mention, '...relevant to the project and theoretical framework'. What is the theoretical framework of this study?
RESULTS:
Line 156 - 158 : 'Approximately half of the staff ...with 157 much less participation from the Surgery department'. The authors stated (Materials and methods) Line 74, '...researchers aimed to sample an entire team of patient-facing staff...'. Can you report the response rate of participation from each depatment in this manuscript? What was the sampling frame (total possibble particpants)
Did you consider 'point of data saturation' as an approach?
My understanding of qualitative research is about exploring the data and not quantifying it. It is about gaining insight and understanding of phenomena. It is subjective. The results must also do justice by those participants who adequately described burnout as per the said definition. One way to represent all perspectives would be to give both accounts: For example, participants indicated experiences of burnout (quotes). On the contrary, there were perceptions related to burnout that were not in agreement with the burnout definitions (quotes)(3.1 adequately discusses this).
Participant 40 (line 192), 'they were dropping me into some night shifts…and I was having palpitations. I could feel it there…like oh god I’m dreading…I was really dreading going in *participant pats their chest during the anecdote*'. Does this not contribute, have a link or could be a manifestation of ''Emotional exhaustion" - (line 41)? I suggest that the authors clearly define the inclusion and exclusion criteria for burnout in the analysis section. Also define parameters/key words for stress.
Was participant 40's description of burnout consistent with the other 68 participants? Were there other themes or perpectives of burnout not reported?
Overall comments:
This manuscript has promise but needs to address the above concerns. I am still not sure how the findings justify the differences in burnout and stress.
Author Response
Please see the attachement.

Reviewer 2 Report
The author/s has chosen an important topic for their paper, the experience of stress and burnout is personal and can have detrimental effects on health. Using a theory lead qualitative approach is theoretically interesting and can give important and interesting insights into the experiences of medical staff. The study also involve a considerable large group of participants. The topic of the study would be of interest to readers of IJERPH. As in any study, there is potential room for improvement. I will outline my concerns below and some suggestions for improvement.
General comments:
- I’m not really sure what is new. The research on stress and burnout in health professionals are rather well established. I’m not saying that the study does not provide new knowledge but the authors need to give a better rational for the study, situating it within the literature and do a better job convincing the readers why this study is needed.
- It is claimed that the study is ethnographic. Usually ethnographic research involves being in the environment where the phenomenon under study takes place, investigating ‘what is going on’ using observations, interviews and documents for example. I’m not sure that ordinary interviews could be accepted as an ethnographic approach but I might be wrong. If so please give a better rational for why and how this I ethnography. This could a great improvement for the study.
- The analysis is ‘thematic’. There are many forms of thematic analysis. Your study would gain from providing some more information about how the analysis were conducted and what where guiding you in this process. It appears at is theory lead, if it is theory lead thematic analysis (see Braun & Clark, 2006) or some other form deductive analysis (see Patton, 2002 for another good read) would be beneficial to describe this in the method section.
- There are some descriptions of how the coding was done, with two researchers conducting this part of the analysis and a third researcher being some kind external audit or validating the findings. There are many was to do this, providing some more info of what kind of criteria or measures you were using to make a good qualitative study would improve the impression of the study. For a thorough discussion on inter rater reliability and other ways to code data, Miles & Huberman, 1994 is a classic in the field.
- Discussing the potential drawbacks (and positive aspects as well) of using the participants own definitions of burnout could give additional perspective to the study.
References
Braun, V., & Clarke, V. (2006). Using thematic analysis in psychology. Qualitative research in psychology, 3(2), 77-101.
Miles, M. B., & Huberman, A. M. (1994). Qualitative data analysis: An expanded sourcebook. Thousand Oaks. CA.: Sage Publications.
Patton, M. Q. (2002). Qualitative research and evaluation methods. Thousand Oaks. CA.: Sage Publications.
Reviewer 3 Report
It is an interesting research that addresses a common topic in scientific literature but with an original approach. The aim was to explore staff experiences of stress and burnout in a health service with a qualitative methodology. The results are interesting and have important practical implications. In my opinion, the study is well designed, the aim is relevant and the methodology is adequate. However, there are some aspects that should be revised.
- Participants
The authors argue that participants were recruited using a combination of purposive and snowball sampling.
Could the authors provide more information about how they recruited the sample? Was there any reason to choose those two hospitals? Are these hospitals located in the same city? Are they representative of the Australian health service? Are they from rural or urban areas?
Could the authors provide more information about the sociodemographic and professional variables of the participants? For example: age, gender, years in the service, years of professional practice.
- Procedure
What information were participants given about the aims of the study when they were encouraged to participate? What information appeared on the Participant Information and Consent Form? Was it the same information?
It would be convenient to include the approximate duration of the interviews and where were them carried out.
- Results
The authors say that the participants frequently discussed what they perceived to be burnout.
Were there participants who claimed not to experience stress or burnout at work?
Although not the objective of this study, could the authors indicate if there were differences int the results due to gender or different professional and clinical specialty groups, as the literature about burnout traditionally points out?
- Limitations
The authors should also include as limitations the need to control variables than can affect burnout (eg. professional group, gender or years in the health service).
In my opinion, although the results shoud be confirmed by future research, the discussion and conclusions generate interesting ideas, focusing attention on organizational factors as social determinants of health. It is a study with interesting practical implications.
Round 2
Reviewer 1 Report
Thank you for your extensive review. I feel you have addressed issues raised in Review One. As such, I do not have additional comments and wish you the best.
Author Response
Thank you.
Reviewer 2 Report
I commend the authors for their job revising their manuscript. I do however think that the method section need improvement. If you are using an ethongraphic approach initially. Please provide information on how this process were conducted and what kind of data were collected and how was it used? In addition, I do not think you did a thematic analysis as descibed by Braun and Clark. If you did then you need to provide a better description and in line with theis suggestions.
Author Response
Thank you for your comments. Our response is attached below.

Round 3
Reviewer 2 Report
I'm pleased with how the authors have impoved the manuscript. Two minor suggestions.
First, you do not need to state the pages in Braun and Clark (2006), see page 3 line 137-144. I also suggest that you consult any of the references that Braun and Clark use when they suggest quality criteria for qualitative research and add this for example on page 3 in 2.4. Data management and analysis. This is nicely presented in their book (Braun & Clark, 2013). When doing so you make the paper more coherent in the use of method and quality criteria.
